# Imprinting Technology for Effective Sorbent Fabrication: Current State-of-Art and Future Prospects

**DOI:** 10.3390/ma14081850

**Published:** 2021-04-08

**Authors:** Marta Janczura, Piotr Luliński, Monika Sobiech

**Affiliations:** Department of Organic Chemistry, Faculty of Pharmacy, Medical University of Warsaw, Banacha 1, 02-097 Warszawa, Poland; mjanczura@wum.edu.pl (M.J.); monika.sobiech@wum.edu.pl (M.S.)

**Keywords:** molecularly imprinted polymers, ion imprinted polymers, separation, solid phase extraction, dispersive solid phase extraction, magnetic susceptible material, core-shell, nanoparticle

## Abstract

In the last 10 years, we have witnessed an extensive development of instrumental techniques in analytical methods for determination of various molecules and ions at very low concentrations. Nevertheless, the presence of interfering components of complex samples hampered the applicability of new analytical strategies. Thus, additional sample pre-treatment steps were proposed to overcome the problem. Solid sorbents were used for clean-up samples but insufficient selectivity of commercial materials limited their utility. Here, the application of molecularly imprinted polymers (MIPs) or ion-imprinted polymers (IIPs) in the separation processes have recently attracted attention due to their many advantages, such as high selectivity, robustness, and low costs of the fabrication process. Bulk or monoliths, microspheres and core-shell materials, magnetically susceptible and stir-bar imprinted materials are applicable to different modes of solid-phase extraction to determine target analytes and ions in a very complex environment such as blood, urine, soil, or food. The capability to perform a specific separation of enantiomers is a substantial advantage in clinical analysis. The ion-imprinted sorbents gained interest in trace analysis of pollutants in environmental samples. In this review, the current synthetic approaches for the preparation of MIPs and IIPs are comprehensively discussed together with a detailed characterization of respective materials. Furthermore, the use of sorbents in environmental, food, and biomedical analyses will be emphasized to point out current limits and highlight the future prospects for further development in the field.

## 1. Introduction

In the last decade, the technological progress introduced advanced instrumental techniques in the field of analytical chemistry, allowing to qualitatively and quantitatively analyze various compounds of interest. The development of highly sophisticated measuring instruments enabled us to determine molecules and ions at very low concentrations. Within this field, novel analytical strategies were elaborated on, but the sample preparation stages were recognized as the main source of insufficient method validation parameters and a key factor of systematic errors. The presence of interfering components in the complex samples substantially affected analytical performance, limiting capabilities of advanced instruments. Therefore, the primary goals of the sample preparation process are related to the preconcentration of analytes, due to the low analyte levels, and reduction of the matrix effects of the sample components or interfering compounds. Moreover, a decrease of the sample volume, reinforcement of automatization, and elimination of non-environmentally benign solvents are also desired. Here, diverse extraction techniques were implemented into analytical strategies to ensure effectiveness of sample preparation, such as liquid-liquid extraction or solid-phase extraction (SPE). The former technique, mostly based on the utilization of organic solvents, provides liquid-liquid extraction not compatible with green chemistry principles. As a result, the SPE was identified as the separation technique of first choice with many alternative modes, such as solid-phase micro-extration, stir-bar sorptive extraction, or magnetic dispersive solid-phase extraction, among others [1,2,3].

In recent years, despite different methodological approaches, the SPE was one of the frequently chosen extraction techniques utilized in the laboratory practice. Its common application enforced proper standardization measures and resulted in the dynamic increase of commercially available solid sorbents of all adsorption—extraction modes, including normal phase, reverse phase, ion exchange, immune affinity, and mixed-mode. Those materials possess various physicochemical parameters, which allow them to be employed in the separation process of compounds of a different chemical nature [4,5]. Nevertheless, new classes of sorption materials are investigated to improve selectivity toward target analytes, increase sorption capacity, enhance mechanical durability, and make the entire process more environmentally-friendly. Among such new materials are silica-based sorbents, magnetic nanoparticles, and carbon-derived materials, such as graphene or fullerene-related carbon nanotubes, metal-organic frameworks, porous polymers, and biopolymers [6,7]. Silica-based materials are easy to synthesize, providing a rigid material with satisfactory resistance to shrinking and swelling but suffer from limited capability for modification and moderate tolerance to extreme pH values. Magnetic nanoparticles facilitate the sorption process by a simple application of an external magnet, exhibiting good dispersibility and high surface-to-volume area, but their functionalization is necessary to avoid oxidation and agglomeration. On the other hand, carbon-derived materials are characterized by a highly specific surface area, providing satisfactory adsorption capacity and acceptable durability. However, their tendency for irreversible aggregation that diminish sorption behavior negatively affect their application, requiring us to immobilize or to modify their surface. The advantages of highly porous metal-organic frameworks are derived from their porosity, which could be easily tunable as well as from a highly specific surface area, which could be easily modifiable. Finally, biopolymers and eco-friendly materials are characterized by high-binding capacity and hydrophilicity, but suffer from low selectivity.

Up until now, an insufficient selectivity of commercial sorbents was recognized as the most challenging problem that should be solved by polymeric chemists and material engineers. Here, the application of molecularly imprinted polymers (MIPs) or ion imprinted polymers (IIPs) in the separation processes have received much attention. Those materials, called the imprinted sorbents, could be an attractive alternative to commercial sorbents since the template-tailored synthesis during the fabrication process introduces a high specificity to the material [8]. The imprinted sorbents have gained attention of many scientific groups, resulting in a plethora of studies published recently on this topic. Moreover, a set of designed imprinted materials was commercialized by various companies such as Biotage AB, MIP Diagnostics, Polyintell SAS, or Semorex Ltd. On the other hand, Sigma-Aldrich offers selected molecularly imprinted solid phase extraction (MISPE) sorbents under Supel-MIP line. Nevertheless, there is still a number of improvements that may be introduced, such as better managing a non-homogeneous population of adsorption sites, controlling the morphology of the material, and improving the template removal step. To overcome these problems, a proper synthesis optimization and careful characterization of imprinted sorbents is required to reveal their sorption nature.

In this review, a critical revision of various synthetic approaches supported by comprehensive characterization is presented, covering the investigations published predominantly within the period of the last five years. Advantages and disadvantages of various methods will be pointed out, aiming at facilitating the design of efficient, imprinted sorbents for the separation purposes. The principles of the molecular imprinting technology (MIT) and the limits that hampered the widespread application as well as extensive commercialization of those materials will also be discussed.

The excellent reviews summarizing the synthesis and application of MIPs and IIPs in various fields were published previously [8,9,10,11,12,13,14,15,16,17]. The most prominent reviews presented by BelBruno [9] as well as Chen and co-workers [10] describe the principles of the imprinting process and their applications. However, the limitations of the performance and handling of such materials are also emphasized. Cheong and co-workers [11] presented in a review the development of imprinted sorbents in the previous decade. The progress in the application of MIPs as detectors resulted in a large number of reviews that were also published recently [12]. Nevertheless, the most interesting, but also the most challenging area of application of MIPs is related to protein separation, the detection of microorganisms, and the usage of imprinted materials in bioimaging and cell targeting [13,14,15]. Drug delivery vehicles based on the imprinted materials also attracted attention. A few different methods of application of MIPs for ocular, transdermal, or oral administration were described in a review by Lulinski [16]. Finally, Murastugu and co-workers [17] revealed a potential of MIPs as versatile tools for catalysis.

## 2. Overview of the Imprinting Process

The MIPs or IIPs are characterized by the degree of selectivity and specificity due to the presence of specific recognition regions in the polymer net that are formed by the template-tailored (either molecule or ion) synthesis. This synthetic process consists of three stages. First, the pre-polymerization structure or adduct/complex is formed from functional monomers and the templated molecule or ion in the presence of an appropriate solvent. It is a crucial stage, since it determines the formation of the specific recognition regions responsible for the imprinted materials selectivity. In the case of MIPs, this step can be completed either by the covalent or non-covalent approaches, which differ in the type of interactions formed between the template and monomers. The covalent approach is executed via the formation of chemical bonds and, as a result, a new pre-polymerization compound is obtained. In the non-covalently formed pre-polymerization complex, weak molecular interactions, such as hydrogen bonds, ionic forces, π-type interactions, or van der Walls forces are responsible for the stability of the system [8]. In case of IIPs, non-covalent or coordination interactions are involved in the formation of the pre-polymerization complex, usually with the support of a proper ligand or chelator [18]. The following step of the imprinting process comprises of a polymerization reaction, usually in the presence of a cross-linker, to fix the pre-polymerization structure and form a polymeric matrix with the specific recognition regions and the template or ion embedded inside of them. In the last stage, the template or ion is removed from the polymer by physicochemical processes, such as, e.g., hydrolysis reaction (in the covalent approach) or desorption (in the non-covalent approach), yielding a highly cross-linked polymeric matrix with three-dimensional specific recognition regions complementary in term of the molecular volume and geometry of chemical functionalities to the imprinted entity. Those regions are able to adsorb/desorb the template molecule/ion or its structural analogues. The main parameter describing the efficacy of the imprinting process is the imprinting factor (IF). The IF is commonly defined as the ratio of the binding capacity of the template or ion on the MIP or IIP, respectively, to the binding capacity of the template or ion on the reference non-imprinted polymer (NIP). To make this definition valid, the conditions of NIP synthesis have to be identical as MIP or IIP production with the omission of only the addition of the template molecule or ion [8].

Despite the advantages derived from the template-tailored synthesis, a number of critical problems that strongly affect the applicability of imprinted materials can be identified. The heterogeneity in the population of specific recognition regions on the imprinted material surface remains the main challenge here. The origins of heterogeneity could be related to the diversity of the functional monomer and cross-linker interactions with the template or ion, usually when the non-covalent approach is considered. The conformational stability of the template, the dimerization or self-complexation of templates, and the chemical stability of the template, and, for IIPs, the effective ion removal (since such sorbents are primary designed to trace analyses) were identified as the factors affecting the recognition behavior of imprinted materials. It should be noted that even a low level of leaking of post-synthetic ions could affect the analysis, providing false results. Below, the problems related to the MIPs heterogeneity will be discussed in detail.

In an excellent work published by Karlsson and co-workers [19], it was emphasized that the cross-linker forms interactions with the template molecule, providing important factors during the formation of a specific geometry of recognition regions in the MIP. It was also noted that the change in the population of template conformations could be driven by the changes in the local microenvironment of the template, governed by the presence of the cross-linker molecules. As a result, it was suggested that a higher concentration of a functionalized cross-linker could be used to provide highly specific MIPs.

The problem of the template conformational stability was also discussed by Olsson and co-workers [20]. The template-template complexation could result in the heterogeneity of the MIP because of the population of highly specific recognition regions from two molecule complexes. Here, the adsorption relies on the template-template interactions. Moreover, additional complexes could be considered, such as those involving two molecules of templates, interacting with one molecule of the functional monomer acting as a bridge between them.

Another reason underlying the formation of the heterogeneous population of adsorption regions was identified by Martin and co-workers [21], who revealed extraordinary selectivity of a propranolol imprinted polymer toward tamoxifen. The mechanism explaining the behavior of MIP was not discussed, but it was concluded that a rigorous optimization of the design of MIPs should be conducted in order to determine fitting to the targeted analyte. In another example, the structural transformations of the template during the imprinting process could also affect homogeneity of the adsorption regions. Klejn and co-workers [22] investigated the polymer imprinted by 3,3-diindolylmethane. It was observed that the resulting MIP provided high binding capacity toward a structurally related compound, known as indole-3-methanol. Subsequent free radical reactions were responsible for the transformation of 3,3-diindolylmethane (template) to indole-3-methanol, which is the compound that was present in the pre-polymerization system and was imprinted into the resulting polymer matrix. As a consequence, a highly heterogeneous population of adsorption regions was formed. Nevertheless, it should be emphasized that, in most cases, templates are stable during the polymerization process.

In order to manage the heterogeneity of specific adsorption regions of MIPs, the theoretical analyses and experimental modifications were introduced into the optimization process of MIP synthesis. The description of mutual interactions between the components of the pre-polymerization system at the molecular level as well as the understanding of their impact for the polymerization and the microstructure of the specific recognition regions in resulting MIP could provide the means to facilitate the production of advanced materials. For such purposes, the molecular modelling was recognized as a versatile tool, as it allows for an in-silico construction of specific recognition region models, as shown for numerous systems [23]. 

The formation of realistic models of the MIP microstructure and macrostructure is very complex and time-consuming due to the multicomponent nature of analyzed material. To make calculations easier and faster, different kinds of simplifications and assumptions can be applied. In order to build the MIP model, the formation of the pre-polymerization systems should be carried out first, to mimic synthetic protocol. The simplest way to use computational methods in design and analysis of MIPs is the formation of a pre-polymerization complex model consisting of only two molecules known as the template and the functional monomer, and an analysis of interactions and binding energies in the studied system. When different monomers are used, the screening process and the choice of the most appropriate monomer for MIP synthesis could be performed. This methodology was successfully applied by Piletsky and co-workers for the design of MIPs recognizing various analytes [24,25,26]. To create a more complicated but realistic model of a pre-polymerization complex, more monomer molecules were taken into account during the modelling process [27]. The simulations of the pre-polymerization system apart from the template and monomers could also involve cross-linker [28], solvent [29], or both [30] as well as all components of the pre-polymerization mixture (template, monomers, cross-linker, solvent, and initiator) [31,32,33,34]. The analysis of simulation results (mainly interaction energy) of the pre-polymerization system was helpful to plan the synthetic process of MIPs, especially in choosing the most suitable monomer [24], cross-linker [28], or solvent [28]. Additionally, the analysis of interactions in the pre-polymerization systems was crucial to explain the molecular recognition mechanism of MIPs [31,32,33,34]. 

In a more advanced approach to MIPs simulation, the models of MIP-specific recognition regions, and cavity models, can be created [35]. Sobiech and co-workers [36,37,38,39] constructed series of MIP cavity models where the positions of monomers and cross-linkers were fixed and double bonds in vinyl groups were replace by single ones to imitate the polymerization process. Next, the template was removed and the adsorption process of different compounds was simulated prior to the analysis. In more complicated systems, the solvent molecules were taken into account during the modelling and the analysis of the interactions [40,41,42]. Another idea of computational modelling was the formation of a polymeric chain model and studying of the analyte-chain interactions. Here, the polymeric chains were constructed from monomers [43,44,45] or from cross-linkers [46,47]. A new simulation algorithm of a radical polymerization process was described by Cowen and co-workers [48]. The methodology was capable of reproducing MIP in silico with higher accuracy than alternative methods. Results obtained from such theoretical investigations were evaluated, compared, and correlated with experimental data, such as adsorption capacity, IF, or chromatographic results, mostly giving good agreement with them. Thus, a computational study can be a useful method in the design of MIPs by avoiding time-consuming trial and error experimental approaches. Moreover, this approach could help us to understand the origins of the heterogeneous nature of MIPs, providing an instrument to design more effective sorbents.

In order to design an effective imprinted sorbent, the comprehensive analysis of physicochemical data obtained during the isotherm determination of the relationship between the concentration and the adsorption capacity of the material is required. For that purpose, various mathematical models were introduced to provide the explanation for the adsorption behavior. Different isotherm models could be applied to characterize the adsorption process on MIPs. The Langmuir isotherm model assumes that the adsorbent surface is uniform and possesses a finite number of identical sites, while the adsorption occurs in a monolayer on the surface of the adsorbent and adsorbate-adsorbate interactions do not occur [49,50]. The Freundlich isotherm describes a multilayer adsorptive process on the heterogeneous surfaces [49,50]. The Jovanovich model is based on the same assumptions as the Langmuir model with the possible addition of some mechanical contact between the adsorbing and desorbing molecules [51]. The Temkin isotherm model assumes that the heat of adsorption decreases linearly with the coverage due to the interactions between the adsorbate and adsorbent [49,52]. The Dubinin-Radushkevich model gives insight into the biomass porosity as well as the adsorption energy [53]. Previously described models were used to calculate adsorption properties of MIPs [49,50,54,55]. However, simple isotherm models show only a simplified version of the adsorption process. In MIPs, the adsorption behavior could be difficult to fit to simple isotherm models due to the heterogeneity of adsorption sites and their non-uniform distribution [54]. Due to the presence of multiple classes of recognition sites, the complex isotherm models, such as bi-Langmuir [56] or Langmuir-Freundlich [57], were introduced in the analysis of MIP properties.

The experimental efforts to improve the homogeneity of the MIPs focused on designing specific functional monomers, enabling the enhancement of interactions with the template in order to stabilize the pre-polymerization structure [58]. It should be noted that combining both theory and practice could provide satisfactory results during the design of effective MIP based on a new functional monomer in the case of 2-hydroxy-3-(isopropylamino)propyl methacrylate [59]. The monomer was synthesized and effective separation of acidic compounds was shown by experimental data. The results confirmed the paradigm shift, which opened a novel approach to synthesize exclusive functional monomers. Lately, however, there are new perspectives to solve the problem of heterogeneity of MIPs. One of them is the employment of ionic liquids during the synthesis of MIPs [60]. Ionic liquids could serve as functional monomers, co-monomers, or solvents. Those compounds are able to create multiple interactions with the templates, including hydrogen bonds, electrostatic, π-π and ion-exchange interaction, enhancing the stabilization of the pre-polymerization complex, providing the specific recognition regions better defined and ultimately improving the performance of the MIPs. Various studies revealed that both cations and anions of ionic liquids have a significant effect on the imprinting effect.

It should also be pointed out that the polymerization reaction is under kinetic control, affecting the stability of complexes, and, as a consequence, providing heterogeneity to the polymer. The effect of the propagation step on the morphology of resulted imprinted material could be significant, resulting in different pore systems of derived MIPs. In order to overcome these problems, the controlled free radical polymerization can be carried out. Recently, this method gained a lot of attention in the synthesis of MIPs and various approaches have been already reviewed [61].

To sum up, the management of the heterogeneity of MIPs is crucial to improve the analytical performance of the imprinted sorbents. The studies of the heterogeneity of MIPs and the application of advanced tools for the prediction of MIPs properties will result in a design of more effective imprinted sorbents.

## 3. Molecularly or Ion Imprinted Sorbents for Solid Phase Extraction—Synthesis Approaches and Characterization

In this section, synthetic approaches to fabricate molecularly or ion-imprinted sorbents will be discussed together with characterization of obtained materials. Various forms of MIPs together with their composition details, characterization methods, and application in the separation process are summarized in Table 1 at the end of the section.

### 3.1. Bulk, Monoliths, and Mesoporous Imprinted Sorbents

The bulk polymerization technique is the most popular and general approach for preparing imprinted sorbents due to the rapidity and simplicity of this process. In a recent paper, Janczura and co-workers [30] provided insight into the morphology, pore structure, and sorption properties of 4-hydroxy-3-nitrophenylacetic acid (HNPA) imprinted poly(acrylic acid-co-ethylene glycol dimethacrylate) sorbent. HNPA serves as an indicator of 3-nitrotyrosine, a biomarker of pathophysiological states. The purpose of the study was to disclose the relationship between the adsorption behavior and morphology of bulk imprinted material. This result was obtained via a comprehensive structural characterization of material. The micrographs from scanning and transmission electron microscopies revealed that the average particles diameter was ca. 10–20 μm with an orderly extended surface but noticeable differences in the morphology of MIP and NIP were disclosed. The MIP and NIP consisted of small regular entities (of a diameter below 100 nm) that formed agglomerates. The entities that formed the surface of MIP were more densely packed, resulting in a lower number of macropores of higher diameter of ca 250 nm when compared to NIP (Figure 1a,b). The atomic force microscopy confirmed a rough surface of MIP and NIP. The three-dimensional surface geometry of MIP revealed domains of intervals that could correspond to cavities and the domains of closely packed floccules entities (Figure 1c). On the contrary, the three-dimensional surface geometry of NIP was more planar. Next, the X-ray diffraction measurements provided information of the crystalline structure of materials. The diffractograms revealed very similar patterns for both MIP and NIP. The presence of broad peaks between 10° and 20° angle (2θ) indicated an amorphous structure of the sorbent. Additionally, the X-ray electron dispersive spectroscopy analysis was used to reveal the presence of carbon and oxygen atoms in the polymer network of MIP and NIP. Moreover, the analysis of elemental mapping images confirmed the homogeneous distribution of C and O atoms in the MIP and NIP structure of poly(acrylic acid-co-ethylene glycol dimethacrylate) material (Figure 1d,e). The absence of nitrogen atoms in the MIP proved a sufficient template removal from the polymer matrix. The ^13^C CP/MAS NMR technique was used to confirm the structure of the material (Figure 1f). Such a comprehensive characterization of the materials could be helpful to explain the sorption behavior as well as to ensure the structural composition of the resulting MIP. Finally, the porosity data of nitrogen sorption (Brunauer-Emmet-Teller isotherm) revealed that MIP possessed a significantly lower specific surface area than NIP (Figure 1g–i). It was against the previously described findings that MIP’s specific surface area was greater than those of NIP. This phenomenon was explained by the impact of the template molecule for the creation of cavities, resulting with an extension of the surface [62].

A more detailed discussion of the relationship between the pore system of different bulk imprinted polymers and their sorption properties was described by Marć and co-workers [63] who prepared MIPs for the separation of selected polybrominated diphenyl ethers. In this study, it was observed that the specific surface area of MIP was also lower than those of NIP despite the higher adsorption capacity of MIP. The fact was explained by the presence of a significant amount of mesopores that were created because of the presence of the template. The formed mesopores gives a possibility to successfully employ prepared polymer material as a sorption medium in the SPE technique due to the good permeability for organic solvents. However, no clear relationships between the employed type of functional monomer and the average pore diameter or average pore surface area were found. It was concluded that the template molecule was the main factor responsible for the creation of the specific recognition regions and a sufficient mesoporous structure enhanced that effect.

Nevertheless, the obtained bulk polymer material needs to be crushed, ground, and sieved into the desired particle size range. Moreover, the post-synthetic treatment results in irregular particles with wide particle size distribution. The template removal step is often unsatisfactory due to limited transfer of desorption solvent into the pore system of particles. The grinding process results in a destruction of surface-specific recognition regions and the entire process is time consuming, providing low yields of the final material. To overcome problems derived from crushed bulk material, monoliths were proposed due to the simple in situ polymerization process, integrated structures, and higher porosity, providing material high rigidity and permeability. 

In a very interesting study, Zhang and co-workers [64] proposed simultaneous application of two templates, namely naproxen and ketoprofen, which are non-steroidal anti-inflammatory drugs that interacted with the functional monomer, 4-vinylpyridine, ethylene glycol dimethacrylate as a cross-linker with Co^2+^ as a pivot. A ternary mixture of dimethyl formamide, dimethyl sulfoxide, and 1-butyl-3-methylimidazolium tetrafluoroborate was used as the porogen system to obtain monolith with high selectivity. Ionic liquid system enhanced permeability of the MIP monolith. The mercury intrusion/extrusion porosimetry was employed to analyze the pore system of the materials. The results revealed a disordered and heterogeneous macropore system with a fast mass transfer in and out of the mesopores, which helped us to avoid fragmentation of liquid during extrusion. The mercury intrusion/extrusion types H1 and H2 hysteresis loops were observed. The phenomenon was explained by the pore blocking/percolation effect. Moreover, the MIP monolith possessed lower total pore volume and higher bulk density than the NIP monolith. The mesopore system played a crucial role in the specific adsorption. 

It should be highlighted that various forms of monoliths were elaborated, namely pipette-tip monoliths and monolithic discs [65,66]. Sorribes-Soriano and co-workers [65] proposed immobilization of the MIP monolith layer on the surface of polytetrafluoroethylene commercial disks, modified through the conversion of C–F bonds into C–H, formyl, and carboxyl groups. The obtained disk surface with decreased hydrophobicity was functionalized by glycidyl methacrylate prior to the hydrolysis of the epoxide bridge. The subsequent chemical modifications of disks were proven by various characterization methods.

To extend the mesoporous structure of imprinted materials, mesoporous silica carriers, such as FDU-12 or SBA-15, were employed. Those three-dimensional mesoporous materials are characterized by a well-ordered pore structure, which allows the analyte and solution satisfactory diffusion into the material. Rui and co-workers [67] used FDU-12 support to develop imprinted sorbent for extraction of aflatoxins and contaminant mycotoxins that could be found in food products or beverages. The precipitation polymerization was applied to obtain a mesoporous aggregated material. The X-ray electron dispersive spectroscopy was used to prove the coating of the imprinted layer. The structure of materials was also analyzed by X-ray diffraction, showing strong reflection peak in neat FDU-12 in the 2θ region of 22.5°, corresponding to a crystalline structure of the support. The peak position modified by the MIP layer material was unchanged, indicating that the structure of the FDU-12 was essentially retained but its intensity was lower. 

In another example, Zhang and co-workers [68] used mesoporous silica SBA-15 as a carrier of imprinted layer toward analysis of bisphenol A, an endocrine disruptor which is commonly found in food samples. Thermogravimetry of neat SBA-15 and SBA-15 modified with an imprinted layer revealed similar trends as the temperature changed from 100 °C to 800 °C with low and high loss rates, respectively. Nevertheless, imprinted material exhibited larger mass loss than neat SBA-15, resulting from the decomposition and degradation of a grafted, imprinted layer. 

The porosity measurements revealed that the specific surface area of neat SBA-15 was higher than those of modified imprinted material (643 to 374 m^2^ g^−1^, respectively), disclosing the impact of an organic layer on the decrease of the surface area and affecting the pore diameter. The X-ray diffraction results indicated that an ordered mesoporous structure of the SBA-15 was well preserved after coating the imprinted layer. 

A similar approach was proposed by Xu and co-workers [69], who described the modification of the surface of SBA-15 support with azide functionalization prior to alkyne-modified β-cyclodextrin and propargyl amine. The imprinted material was dedicated to analysis of 2,4-dichlorophenoxyacetic acid (2,4-D), which is an herbicide commonly used to control the growth of weeds and broadleaf plants. In the first step, the synthesis of alkyne modified β-cyclodextrin was carried out and the comprehensive analysis of intermolecular interactions with template of 2,4-D was provided, employing ^1^H NMR spectroscopy. It allowed the authors to determine the structural regions of the template, which interacted with the host monomer. Here, the shifts of aromatic protons were most noticeable, confirming that the 4-D was inserted into the cavity of β-cyclodextrin. The nitrogen sorption analysis revealed that the isotherms of neat SBA-15 and SBA-15 modified MIP could be classified both as the type IV isotherm, typical for mesoporous solids with uniform size distributions. 

To sum up, the substantial progress in the synthetic approaches and proposed forms of imprinted sorbent was identified recently, allowing new materials effectiveness in the separation process. Nevertheless, the shortcomings derived from surface imperfections of bulk and monolith materials or from incomplete modification of silica supports oriented the investigation into the polymeric microspheres that could be obtained by the precipitation or emulsion polymerizations, multistep swelling polymerization, or fabrication of more sophisticated, core-shell particles.

### 3.2. Microspheres and Core-Shell Imprinted Sorbents

The large variability and irregularity in the particle size is not beneficial for sorbent applications of imprinted materials, but it can be solved by the use of precipitation polymerization, which leads to uniform morphology of materials. Neolaka and co-workers [70] synthesized poly(4-vinylpyridine-co-ethylene glycol dimethacrylate) polymer imprinted by potassium dichromate for environmental pollutant extraction. The X-ray diffraction measurements of IIP and NIP revealed an amorphous structure of materials. However, the analysis of IIP before the ion removal revealed the presence of distinct peaks at 2θ = 24.38°, 25.65°, 27.07°, 29.76°, and 31.12°. The X-ray electron dispersive spectroscopy confirmed the presence of Cr(VI) ions at 1.21 wt% and at 0.68 wt% in the IIP before and after ion removal step, respectively. It was concluded that the ion removal step was insufficient, reaching only 60% of effectiveness due to high rigidity of the material and strong interactions between ions and specific recognition regions.

Zeng and co-workers [71] designed MIP for purification of tylosin, a macrolide antibiotic extracted from the fermentation broth of *Streptomyces fradiae*, by a two-step seeded precipitation polymerization process. In the first step, seed emulsion of poly(methacrylic acid-co-ethylene glycol dimethacrylate) was obtained, which was followed by the second step of the formation of an imprinted layer. Slight differences in the specific surface area were explained by the compactness derived from the interaction between the template and monomers during the self-assembly process and the impact of the template on particle growth during the precipitation polymerization. 

Nevertheless, the imprinted sorbents obtained in previously mentioned studies suffered from a high degree of agglomeration of particles. Thus, the strategies to synthesize microsphere forms of imprinted sorbents could be advantageous because of a regular spherical shape and monodispersity of particles. Those properties resulted in better separation performance of imprinted materials as sorbents. In this case, mostly the emulsion polymerization technique was utilized. This polymerization method enabled us to control the microspheres size distribution since the size depends on the nature of the dispersed and continuous phases.

Pourfarzib and co-workers [72] used a mini-emulsion polymerization technique to obtain a sorbent for selective extraction and purification of efavirenz, a non-nucleoside reverse transcriptase inhibitor, from human urine and serum. The polymer was synthesized in the presence of hexadecane, acting as a hydrophobic agent and sodium dodecyl sulfate as the surfactant. To analyze the structure of the sorbent, infra-red spectroscopy was employed, indicating similar and typical peaks for both materials, known as MIP and NIP. However, variations were observed between MIP samples recorded before and after the template removal step. The lower electric cloud density of –OH and –C=O stretching vibrations at 2966 and 1733 cm^−1^ in the infra-red spectra proved the template removal from the matrix. The material characterization included scanning electron microscopy and photon correlation spectroscopy. It was revealed that MIP possessed a higher diameter of microspheres (253 ± 40 nm) and lower polydispersity index (0.20) when compared to NIP (185 ± 35 nm and 0.32, respectively). In the molecular imprinting process, the nucleation and growth explain the distinctions of the size and size dispersibility between MIP and NIP even under the same synthesis conditions. Nevertheless, the microspheres were polydisperse and the particle size analysis obtained by photon correlation spectroscopy revealed a variation between 217 and 253 nm.

The diameters of polymer microspheres prepared by the mini-emulsion polymerization were of a few hundred nanometers, which are more suitable for applications in liquid chromatography separation rather than MISPE. Thus, the development of alternative techniques for producing MIPs with suitable particle sizes was needed. To solve these problems, Zhang and co-workers [73] proposed application of Pickering emulsion polymerization for matrix solid-phase dispersion extraction of antifungal active ingredients in personal care products, which could be found in sewage. The size of the imprinted microspheres was carefully optimized by changing the high-shear dispersing time and speed, obtaining particles in the range of 20–50 μm. The Pickering emulsion was then formed by mixing the pre-polymerization mixture and aqueous dispersion phase. The optimized synthetic process allowed us to tune the diameter of particles. 

A similar approach was proposed by Yang and co-workers [74] to obtain selective imprinted sorbent for the SPE of eight bisphenols from human urine samples. The synthetic step differs from a previously described paper by using 4-vinylpyridine as the functional monomer. The amount of siloxane particles on formation of polymer microspheres was investigated, revealing the demulsification of the prepared Pickering emulsion existing in a low mass of siloxane. Additionally, the effect of surfactant volume on the stabilization of emulsion in the presence of the template was analyzed, revealing that Triton X-100 could act as a co-stabilizer with siloxane nanoparticles to eliminate the negative effect of template on oil/water phase stability. The porosity of MIP and NIP were also investigated and both materials showed a type IV nitrogen adsorption-desorption isotherm, which is associated with the capillary condensation in the mesoporous materials. 

Cao and co-workers [75] described the polymer imprinted by the resveratrol, the E-isomeric derivative of stilbene, on the surface of silanized porous cellulose microspheres covered by an organic copolymer layer from 4-vinylpyridine (functional monomer) and ethylene glycol dimethacrylate (cross-linker). The support of porous cellulose microspheres are characterized as biodegradable, biocompatible, and green cores with a larger specific surface area than other cellulose materials. The functionalization was carried out using 3-methacryloxypropyltrimethoxysilane. The material was characterized by infra-red spectroscopy, revealing the presence of characteristic peaks attributed to the stretching vibration of –O-H in the cellulose. The X-ray diffraction measurements revealed two peaks at 2θ = 20.2° and 22.5° from cellulose crystals. It was confirmed that the crystal structure of cellulose matrix was completely reserved during the preparation process. Interestingly, ^1^H and ^13^C NMR spectroscopy were employed to prove the progress of the template removal step. In the ^1^H NMR spectrum, the chemical shifts after elution showed variations at 4.59 ppm of H atom form the cross-linker and 1.10 ppm of the H atom from the monomer.

Finally, Wang and co-workers [76] used Pickering emulsion polymerization to obtain molecularly imprinted multicore rattle-type microspheres. The siloxane particles were used as the only stabilizer while hexadecane was used as the solvent. The morphology analyses using scanning electron microscopy revealed the void inside the spheres that was induced by the polymerization-driven phase separation approach. The template molecule of bisphenol A was found to serve as a heterogeneous nucleus, which superimposes on the polymerization-induced phase separation, creating the opportunities for the formation of the multicores encapsulated in the shell (Figure 2a–e).

The presence of surfactants or other agents that stabilize the emulsion is necessary to avoid the collision of oil droplets and Oswald ripening. However, the existence of additional components of the pre-polymerization mixture could affect the imprinting efficacy and could also be a source of heterogeneity. For that purpose, the core-shell microspheres were recognized as the attractive alternative.

The synthesis of core-shell imprinted materials allowed for the specific recognition regions to be located at the surface layer to produce a fast and easy sorption process of analyte in the polymer net. In general, organic, inorganic, or hybrid organic-inorganic cores were employed to serve as support for the external imprinted layer. Negarian and co-workers [77] synthesized core–shell molecularly imprinted particles as the sorbent in the SPE of lincomycin, a lincosamide antibiotic, frequently overexploited for dairy cattle and feed or drinking water for poultry. The core was synthesized from methacrylamide cross-linked by ethylene glycol dimethacrylate and the imprinted shell layer consisted of acrylamide and ethylene glycol dimethacrylate imprinted by lincomycin. 

In another example of using an organic core, Qian and co-workers [78] used glucose as a source of the carbon in hydrothermal carbonization in the presence of acrylic acid prior to functionalization by poly(1-vinyl-3-sulfopropylimidazolium). Then, the lysozyme (the template) was immobilized at the surface and dopamine was polymerized to form a polydopamine shell prior to the template removal. The surface elemental analysis was executed by applying X-ray photoelectron spectroscopy. The results revealed that only carbon and oxygen atoms were present in neat core, while core-shell imprinted material possessed the new peaks that belong to nitrogen and sulfur atoms, respectively. Thus, the polydopamine layer was successfully formed on the surface of the core.

In another approach, Mohiuddin and co-workers [79] proposed the use of a hybrid organic-inorganic core that was composed of polystyrene-coated siloxane. After coating the siloxane layer at the surface, the removal of the polystyrene core was carried out, resulting in a porous siloxane sphere that was used as the rough surface support for the shell layer imprinted by carbamazepine, an antiepileptic, and a mood stabilizing drug. The obtained material was characterized by significant improvement of the surface-to-volume ratio, enhancing the sorption behavior.

In many examples of similar studies, the core consisted of siloxane. Zhu and co-workers [80] designed siloxane-based core-shell imprinted sorbent for the simultaneous analysis of eighteen amino acids in tobacco and tobacco smoke. The siloxane core was functionalized by vinyltrimethoxysilane prior to the polymerization of the layer of poly(methacrylic acid-co-ethylene glycol dimethacrylate) imprinted by the amino acid, theanine. The results from morphology analysis revealed a highly spherical and smooth surface of imprinted core-shell material, suggesting that the organic imprinted layer resulted from the highly selective polymerization of monomers at the surface of silica microspheres. The particles were approximately 300 nm in size and relatively monodisperse. The thickness of the imprinted layer was approximately 40 nm.

Recently, alternative core-shell composites were proposed to extend the effectiveness of the imprinting process or to enhance the structural stability of MIPs, their high specific surface area, and improved anti-fouling performance. Wu and co-workers [81] described the synthesis of TiO_2_ core as a support for the polydopamine imprinted shell to form a nanocomposite for separation of artemisinin, which is the high-efficiency antimalarial drug. The modified multi-level nanocomposite microstructures on the surfaces of nanospheres were observed using scanning electron microscopy. The TiO_2_-modified layers functionalized by 3-methacryloxypropyltrimethoxysilane and poly(vinylidene fluoride) with uniform thickness were observed on the surfaces of polydopamine nanospheres and porous nanocomposite, obtained in the phase inversion process (Figure 3a–i). The elemental mapping from X-ray electron dispersive spectroscopy confirmed the composition of the composite (Figure 3(i1–i5)).

In another work, Guo and co-workers [82] described the synthesis of imprinted sorbent for luteolin, a flavonoid that plays an important role in therapy of various diseases. For that purpose, the core of SiO_2_ as the substrate coated with ZrO_2_ and 3-aminophenylboric acid was used as the boronate affinity functional monomer prior to a thin imprinting layer, which was formed on the polymer surface with polydopamine. Although the material was characterized by a satisfactory adsorbability, selectivity, and reusability, a time-consuming fabrication step was recognized as disadvantageous.

Finally, Wei and co-workers [83] proposed a core-shell organic-inorganic hybrid MIP based on metal-organic framework (MOF-177), known as a material of open crystalline frameworks with permanent porosity, that is characterized by a large surface area, good thermal stability, and cubic cavities of uniform size. Here, the MOF-177 was used as a support for the imprinted layer that consisted of methacrylic acid and tetraethoxysilane with S-amlodipine, a calcium channel blocker administrated in high blood pressure or coronary artery disease, acting as the template. The material was comprehensively analyzed when employing thermogravimetry and scanning electron microscopy. Moreover, the nitrogen sorption data revealed a huge difference between neat MOF-177 and material modified by the imprinted layer. Interestingly, very high specific surface area of neat MOF-177 of 3377 m^2^ g^−1^ was reduced to only 29 or 32 m^2^ g^−1^ after MIP or NIP modification, respectively. The isotherm of MOF-177 of nitrogen sorption was classified as type I. On the contrary, the isotherms of MIP and NIP modified materials with characteristic hysteresis loops were classified as type IV. The modification of MOF-177 resulted with a total change of pore structure from micropores to mesopores and macropores. Nevertheless, the satisfactory adsorption capacity has shown that the mesopore system of the MIP was likely responsible for the specific adsorption behavior. 

To sum up, micro-spherical and core-shell MIPs possess a better adsorption capability than traditional forms because of a high surface-to-volume ratio and a relatively small size.

### 3.3. Magnetic Susceptible and Stir-Bar Imprinted Conjugates

The utilization of sorbents with magnetic properties may facilitate the separation process. The magnetic materials possess plenty of advantages over conventional solid sorbents, such as the high surface-to-volume ratio, fast and effective adsorption of target analytes, simplicity, short extraction times, and low costs. Moreover, the utilization of magnetic sorbents in the dispersive mode of SPE facilitates separation and makes analytical protocols more convenient.

Most of the studies in this area focus on the synthesis and characterization of magnetic core-shell materials, but there are also some works devoted to molecularly imprinted stir-bars. Sobiech and co-workers [84] proposed a magnetic core conjugate with an imprinted shell for sorption of hordenine from human plasma for pharmacokinetic study. Here, the functionalization of magnetic core was proceeded with 3-(trimethoxysilyl)propyl methacrylate prior to the polymerization in the presence of N,N-dimethylphenethylamine, a template, and a structural analog of the target analyte known as hordenine. Application of structural analogs of target analytes as templates during synthesis of MIPs is beneficial since it allows overcoming the template leakage. This process occurs due to the insufficient template removal step and it could affect the results of analytical performance, particularly during trace analyses, making the method unusable. The scanning electron micrographs revealed that the nanoparticles of magnetite, Fe_3_O_4_, were characterized by a fairly uniform spherical shape, but were also agglomerated, likely due to their magnetic nature. The external imprinted shell was characterized by numerous entities uniformly coated by the organic layer (Figure 4a–e). The diameter of magnetite Fe_3_O_4_ ranged between 31 and 53 nm. The thickness of the silane layer conjugated to magnetite particles was between 5 and 8 nm while the thickness of the imprinted shell was between 18 and 31 nm (Figure 4f). The nitrogen sorption analysis disclosed that the magnetite material revealed the hysteresis loop that could be assigned to the H1 type and could refer to the porous materials that form agglomerates. On the other hand, silanized magnetite as well as final core-imprinted shell material revealed hysteresis loops that refer to H3-type or intermediate H3/H4-type characteristics for slit-shaped pores. The specific surface area of magnetite was equal to 34 m^2^ g^−1^. It decreased for silanized core to 7 m^2^ g^−1^ but increased further for a core-imprinted shell to 19 m^2^ g^−1^ (Figure 4g–i).

X-ray diffractograms revealed six peaks observed at 2θ equal to 30.3°, 35.7°, 43.3°, 53.7°, 57.4°, and 62.6°, which can be indexed as (220), (311), (400), (422), (511), and (440), respectively. The results showed that the crystalline structure of magnetite (Fe_3_O_4_) or maghemite (γ-Fe_3_O_4_) was well preserved after further covering of silane and molecularly imprinted layers. However, X-ray diffractometry analysis could not discriminate between magnetite or its oxidation product, maghemite, because the patterns overlapped. Finally, the vibrating sample magnetometer analysis was employed to study the magnetic characterization of magnetite, silanized core, and imprinted as well as non-imprinted shell material. The saturation magnetization was the highest for neat magnetite. The value of magnetization halved after silanization and decreased further after coating with imprinted or non-imprinted organic shell, while still providing values of 15 and 23 emu g^−1^ for MIP and NIP material, respectively. This fact confirmed the conjugation of the external layer (Figure 4k). Nevertheless, the crucial element during the synthesis of magnetic core-shell materials that could affect superparamagnetic properties and its adsorption behavior is related to the control of the thickness of the organic imprinted shell and its porosity.

For that purpose, Liu and co-workers [85] proposed mesoporous SBA-15 material as a support for magnetite nanoparticles. In order to control the thickness of the organic shell, the controlled polymerization was utilized. The reversible addition-fragmentation chain transfer polymerization was employed to polymerize acrylamide (functional monomer) and N,N-methylenebisacrylamide (cross-linker) in the presence of cerium ions. The adsorption data of cerium ions on the material in different temperatures were fitted into the various mathematical models: Langmuir, Freundlich, and Redlich-Peterson, giving the highest correlation coefficient for the latter one. It was found that the adsorption capacity of cerium ions on imprinted and non-imprinted materials increased with higher temperature values, suggesting that heat could provide ample power for metal ions to cross the boundary into ion-specific regions in the polymer net. Moreover, the adsorption capacity of cerium ions was higher for IIP than for NIP at the same temperature. To study the kinetics of adsorption, the Thomas equation and the Adams-Bohart equation were chosen to investigate the process. It was observed that the kinetics of dynamic sorption process on material obtained by controlled polymerization was faster than those for an uncontrollable one. It was generalized that the dynamic adsorption reached equilibrium in a shorter time under the conditions of the greater velocity, higher temperature, and greater initial concentration of cerium ions. The phenomenon was explained by high velocity, high temperature, and high ion concentration, which are the parameters that mostly contributed to the number of crossed Ce(III) ions per unit time, increasing the opportunities for cerium ions to contact with adsorbent and providing shorter time for the adsorption equilibrium.

In order to increase the adsorption capacity, Zhao and co-workers [86] proposed to use magnetic core-imprinted shell particles containing graphene oxide. Graphene oxide is a two-dimensional lamellar structure with a highly extended specific surface area (ca. 2600 m^2^ g^−1^). It could be attached to the magnetic core by physical adsorption or chemical bonding. In the latter case, the functionalization of the magnetic core is compulsory. In the work of Zhao, hexane-1,6-diamine was used to functionalize the surface of magnetite. The organic external layer consisted from acrylamide and ethylene glycol dimethacrylate imprinted by 3-phenoxybenzoic acid as the template. The results from porosity analysis indicated that, despite the utilization of highly surface extended materials, the resulting imprinted core-shell polymer possessed a relatively, weakly enlarged adsorption surface. Nevertheless, the pore volume of MIP was nearly three-fold higher than those for NIP. 

In another approach to utilize carbon materials for surface extension of the resulting magnetic core imprinted shell, Li and co-workers [87] described the employment of magnetic carbon nanotubes functionalized by carboxyl groups for coating of the siloxane layer imprinted by aristolochic acid as the template. Finally, Yao and co-workers [88] described more sophisticated, cage-structured forms of magnetic, imprinted particles. Those single hole-hollow materials are extremely interesting for the purpose of separation because of the highly specific adsorption surface and the ease of modification. Here, Janus-like magnetic, molecularly imprinted nano-bottles were designed by the repeated modification of the bottle-like particles with 3-chloropropyl groups capped on the exterior surface and amino-groups capped on the interior surface, for the separation of luteolin, which is a flavone with potential antioxidant, antitumorigenic, and antiallergic activities. First, silica nano-bottles derived from anisotropic emulsion containing silane coupling agents, such as 3-aminopropyltriethoxysilane, 3-chloropropyltrimethoxysilane, and tetraethyl orthosilicate, were prepared prior to the subsequent modification by magnetic nanoparticles using thermal decomposition of ferric triacetylacetonate, which were attached to carboxylic acid functionalized interior surface of nano-bottles. The organic imprinted polymers, using 4-vinylphenylboronic acid as a functional monomer, were introduced onto the exterior surface of magnetic nano-bottles by an atom-transfer radical polymerization strategy. This allowed us to control the polymerization process, making the organic layer thin. The obtained material was characterized by the hydrophilic nature from hydroxy groups that were present on the surface from 4-vinylphenylboronic acid residues. As a result, the improvement of the separation performance in an aqueous environment was achieved, making the material proper to analysis of biomedical samples.

It should be underlined that, until now, the selected magnetic molecularly imprinted polymers were commercialized. For the purpose of separation of a group of sixteen polycyclic aromatic hydrocarbons in the dispersive SPE, a commercially available magnetic sorbent, provided by NanoMyP^®^ (Granada, Spain), was used by Villar-Navarro and co-workers [89]. The commercial material was characterized as microparticles with a diameter of 3 μm, containing 5% w/w of magnetite with the saturation magnetization equal to 1.52 emu g^−1^ and a specific surface area of 30 m^2^ g^−1^. The elaboration of the analytical method for the purpose of a simultaneous determination of a compound of different physicochemical properties could be challenging. Thus, the comprehensive optimization of the separation process was carried out, beside the fact that the general protocol of extraction was provided by the manufacturer’s guideline.

Finally, the magnetic stir-bar sorbents for stir-bar sorptive extraction were proposed. The application of this technique fulfils the requirements of green chemistry, because it significantly reduces the use of toxic solvents as well as provides the automation of the process by eliminating the time-consuming and laborious preparation steps. As an example, Diaz-Alvarez and co-workers [90] proposed the entrapment of modified magnetic nanoparticles in the imprinted polymer monolith. Here, it was revealed that the percentage of magnetite could affect the effectiveness of the imprinted sorbent. The percentage of magnetite nanoparticles above 28% resulted in no polymerization. On the contrary, the percentage of magnetite below 3.5% resulted in a stir-bar, which did not respond to a magnetic field, making it useless for stir-bar sorption. It was also noted that a high amount of magnetite nanoparticles disrupted the formation of the polymer, affecting the mechanical stability of the obtained stir-bars, making further improvements necessary.

Recently, a number of extensive studies of magnetic imprinted polymers were carried out to provide an even more detailed description of this class of very interesting materials [91,92,93].

### 3.4. Miscellaneous

Apart from the previously described forms of molecularly imprinted sorbents, a few interesting miscellaneous approaches for advanced, imprinted sorbents were described in literature. De Middeleer and co-workers [94] proposed MIPs immobilized on three-dimensional printed scaffolds as novel SPE sorbent for metergoline. Here, the MIP was fabricated as the precipitated polymer build up from methacrylic acid (functional monomer) and trimethylolpropane trimethacrylate (cross-linker) prior to immobilization on poly-ε-caprolactone support functionalized by a Pluronic^®^ F127 bismethacrylate polymer network building block. The material was characterized by microCT analysis to reveal the thickness after each subsequent step of fabrication (Figure 5).

Among miscellaneous forms of MIPs, one can also find fibers, which can be used for solid phase microextraction of various compounds. For example, Demirkurt and co-workers [95] described merging of electrospinning and MIP to obtain an efficient sorbent system dedicated for analysis of paraben derivatives from food and cosmetic products. The synthetic approach involved the preparation of spherical MIPs composed from methacrylic acid (functional monomer) and trimethylolpropane trimethacrylate (cross-linker) that were placed together with 10% (*w*/*v*) polystyrene solution in N,N-dimethylformamide prior to electrospinning on bare silica fibers. The resulting material was characterized by scanning electron microscopy, which showed that electrospinning with polystyrene proceeded well and produced thin, long structures on the surface of silica fiber. Higher magnification revealed the presence of imprinted microspheres incorporated into the fiber structure. The extraction efficacy of new material was compared with commercial fibers, showing that, under optimized conditions, the prepared MIP-coated fiber showed better extraction ability than all the others.

Finally, Dai and co-workers [96] described fabrication of microporous chitosan derived foam for selective biosorption of uranium ions. The structure of the cross-section of material exhibits a homogeneous honeycomb-like porous structure with two classes of pore size equal to 80–100 μm and 5–10 μm, detected on the sheet surface (Figure 6a–d). The elemental mapping showed the uniform distribution of C, O, N, and U after the loading step. The specific surface area was equal to 224 m^2^ g^−1^, an apparent density of 115 mg cm^−3^, a porosity ratio of 93%, and a pore volume equal to 8.1 g cm^−3^ (Figure 6(e1–e4)).

The X-ray photoelectron spectroscopy were employed to investigate the mechanism of the uranium ions adsorption. It was suggested that the amine and hydroxyl groups were responsible for the adsorption, but the contribution of hydroxyl groups was less emphasized. The advantage of the macroporous structure of the material laid in the fast adsorption but limited selectivity in recognizing uranium, actinide, and lanthanide ions was disadvantageous. This fact was explained by the high adsorption energy for 5f/6d orbital of actinides and 4f orbital of lanthanides. In conclusion, it was noted that the fast kinetic, excellent selectivity and facile recovery provided by novel material paved the way for the separation of uranium ions from nuclear industrial wastewater.

## 4. Molecularly Imprinted Sorbents for Enantio-Separation

The chiral separation is MIP’s substantial advantage, which is derived from the geometry of the template molecule in the template-tailored synthesis. The high stereoselectivity is extremely important particularly when considering drug analysis, since the majority of the drugs possess chiral atoms and their pharmacological activity is exhibited only by one isomer, or only one isomer shows a significantly higher activity. Thus, the capability of MIPs to perform chiral separation, mainly as chiral stationary phases, has been a topic of many investigations. It also should be noted that the chiral imprinted sorbent for SPE gained a lot of attention recently, resulting in an excellent review by Moein [97].

## 5. Application Potential of Imprinted Sorbents 

The current challenges of the separation process are related to the insufficient selectivity of commercial sorbents, low preconcentration capabilities, and unsatisfactory elimination of the sample components or reduction of the matrix effect. The main areas of the analytical methods where molecularly imprinted sorbent were used correspond to environmental, food, and biomedical or clinical analyses. The analytical methods devoted to analysis of environmental pollutants, such as ions at very low concentrations, require selective sorbents characterized by high enrichment factors. Additionally, the complexity and diversity of food samples generate problems during the determination of selected food ingredients. On the other hand, the biomedical samples are characterized by a high degree of individual variability and complexity, regarding the tissue or biological fluids of their origin. In view of the potential problems in order to fulfill the current demands from analytical methods of various kind of samples, molecularly imprinted sorbent could be a very promising material. Since the number of studies in this area, even the most recent ones, is vast, we describe below only the selected examples highlighting the application of molecularly imprinted sorbents in the fields of environmental, food, and biomedical analyses.

Zhang and co-workers [98] described the analysis of aminoglycoside antibiotics in the environmental samples. Those antibiotics are frequently used and overdosed in the medical field to treat bacterial infections of humans, but, in many cases, their elimination from the organism occurs without any metabolism. For example, streptomycin elimination is mainly renal with 50–60% being excreted by kindeys. As a consequence, many antibiotics are present in sewage, generating potential risk of antibiotic resistance in ecosystems. In order to monitor the levels of antibiotics in the environment, a novel and selective analytical method based on molecularly imprinted sorbent was proposed. The sorbent was utilized in the optimized SPE coupled with the hydrophilic interaction-high performance liquid chromatography-tandem mass spectrometry. The selectivity of sorbent was analyzed toward a group of the following compounds: streptomycin, kanamycin, apramycin, gentamycin, tobramycin, paromomycin, and spectinomycin, showing high cross-selectivity and similar recovery of all tested analytes, except for spectinomycin. The developed analytical strategy revealed good correlation for all the analytes and satisfactory recovery values ranging between 71% and 108%. The relative standard deviations varied from 2.6% to 11.4% at the three different concentration levels in spiked water samples. The limit of detection ranged from 0.006 to 0.6 µg L^−1^. It was summarized that sorbent possessed great application potential for detecting residual aminoglycoside antibiotics in other matrices. 

The molecularly imprinted sorbents were also applied in the environmental analysis for determination of estrogens in water samples [99], chlorpyrifos [100], triclosan [101], polycyclic aromatic sulfur heterocycles in seawater [102], quinoline [103], and phenolic compounds [104] in cooking water, antibiotics such as norfloxacin [105], levofloxacin [106], erythromycin [107], steroid hormones from water [108], metronidazole [109], rosuvastatin [110], carbamazepine [111], sulfonamides, and their acetylated metabolites from environmental water [112], triazines such as cyromazine from seawater [113], phenols and phenoxyacids from contaminated waters [114], phthalate esters [115], fenoprofen from wastewater [116], chlorsulfuron from soil [117], and various ions, such as mercury (II) [118] or copper [119].

Yuan and co-workers [120] proposed magnetic, molecularly imprinted, sorbent-based, multi-walled carbon nanotubes for simultaneous selective extraction and analysis of a group of phenoxycarboxylic acid derived herbicides in cereals. Those compounds were frequently used in farmlands to increase the yields of cereals. However, the residual presence of its derivates in food might affect the human endocrine system, if consumed. In order to control the levels of phenoxycarboxylic acid-derived herbicides in food, the maximum residual limits (MRL) were introduced in many countries. For example, the European Food Safety Authority stipulated that the MRL of 2,4-dichlorophenoxyacetic acid is 100 ng g^−1^ in rice and 50 ng g^−1^ in millet, oat, and barley or the MRL of 2-methyl-4-chlorophenoxy acetic acid is 50 ng g^−1^ in rice and millet and 200 ng g^−1^ in oat and barley. Thus, a new analytical method was proposed to control the MRL in the cereals using the previously mentioned sorbent. First, the matrix effect was investigated. For complex matrices, the matrix effect is unavoidable in quantitative detection of trace analytes by mass spectrometry mainly for electrospray ion sources. The significant majority of tested analytes, namely 2-methyl-4-chlorophenoxyacetic acid, 4-chlorophenoxyacetic acid, 2,4-dichlorophenoxyacetic acid, and 2-(2,4-dichlorophenoxy)propionic acid showed signal suppression in analyzed cereals, such as rice, millet, oat, and barley. It means that the matrix effect could reveal a substantial impact on the analytical performance, affecting the limits of detections of the method. The only exception was observed for 2-methyl-4-chlorophenoxyacetic acid in barley and 2,4-dichlorophenoxyacetic acid in millet. However, it was demonstrated that the matrix effect values for most analytes were within ± 20%, which was close to the relative standard deviation values for repeatability and could be considered as a weak interference. Nevertheless, for 4-chlorophenoxyacetic acid and 2,4-dichlorophenoxyacetic acid in rice samples, the matrix effect was significant. To eliminate the effects of matrix interference and to ensure satisfactory accuracy and sensitivity of the method, matrix matching standard curves were employed. It allowed us to obtain good linearity in the range of 5–250 ng g^−1^ with a correlation coefficient of 0.9933–0.9992 for matrix matching standard curves of all tested herbicides in cereal matrices. The limit of detection and quantification for target analytes were within the range of 0.33–1.50 ng g^−1^ and 1.25–5 ng g^−1^, which is satisfactory with respect to MRL. The recoveries for all tested herbicides in different cereals ranged from 86.7% to 95.2% with intra-day and inter-day precision not higher than 8.5% and 10.6%, respectively. Finally, the method was successfully verified by an application to real matrices. The obtained extracts, under optimal magnetic SPE conditions, were detected by ultra-performance liquid chromatography combined with tandem mass spectrometry, showing many advantages such as a simple operation, high accuracy, satisfactory sensitivity, and low cost.

The molecularly imprinted sorbents were utilized in the food analysis for determination of antibiotics such as norfloxacin [121] or cephalexin in pork [122], tetracycline in chicken [123], aminoglycosides (streptomycin, kanamycin, and gentamycin) in various milk samples [124], chloramphenicol in honey [125], zearalenone in wheat [126], estrogens (estrone, estriol, and estragon) in milk [127] or in cucumber, milk powder and grass carp samples [128], sulfonamides such as sulfamethoxazole in milk [129], imidacloprid [130] or kaempherol in apples [131], patulin in apple juice [132], hesperidin in lime juice [133], strobilurin in peach [134], carbendazim in orange [135], dopamine in bananas [136], fenoxycarb in mussels [137], phenylarsonic compounds in chicken and pork samples [138], or organochlorine fungicides in ginseng samples [139], acrylamide in biscuits [140], bisphenoles A [141] as well as F and S (on imprinted commercial sorbent—Affinimip^®^) [142], and quercetin in onion [143]. The imprinted sorbents were also used for selective extraction of plant ingredients, such as rosmarinic acid from *Rosmarinus officinalis* L. [144], tannins from the barks of *Anadenanthera macrocarpa* var. *cebil* (Griseb.) Altschul, *Myrciaria jabuticaba* (Vell.) and *Spondias tuberosa* Arruda [145], polydatin from the root of *Polygonum cuspidatum* Sieb. et Zucc. [146], quinolizidine alkaloids from *Sophora flavescens* Aiton [147], or *Eucommia ulmoides* (Oliver) [148]. Those sorbents were utilized to control the dietary supplements ingredients [149].

Finally, as an example of the application of MIPs for biomedical analysis, Jouyban and co-workers [150] proposed imprinted sorbent for the clinical pharmacokinetic study of the valproic acid (an anti-epileptic drug) in the exhaled breath condensate sample. Such samples are preferred due to the non-invasive nature of their collection. However, the sample is very complex, containing different chemical compounds such as volatile or non-volatile biomolecules, proteins, lipids, and carbohydrates in addition to valproic acid. Here, a sorbent was utilized in the SPE method combined with a deep eutectic solvents-dispersive liquid-liquid microextraction prior to gas chromatography-mass spectrometry of the analyte. A careful optimization of the SPE process was carried out in terms of type and volume of the eluent as well as the flow rate and salt addition prior to optimization of dispersive liquid-liquid extractions protocol, employing deep eutectic solvent. The analytical method provided low limits of detection and quantification of 0.04 and 0.13 µg L^−1^ as well as 0.08 and 0.26 µg L^−1^ in the deionized water and the exhaled breath condensate, respectively. The method had proper repeatability, accuracy, and stability expressed as relative standard deviations less than 4.9%, 13%, and 18%, respectively. The valproic acid concentration in collected samples was determined by the method in the 67–152 µg L^−1^ range. Finally, the greenness of the developed method was investigated by considering the analytical eco-scale. In conclusion, it was stated that the analytical method could be applied in bio-equivalency investigations in the pharmaceutical industry.

The molecularly imprinted sorbents found an application in biomedical or clinical investigations of the levels of various important biomolecules, such as biogenic amines dopamine in human urine [151], tryptamine in human cerebrospinal fluid [152], psychoactive compounds such as cocaine [153] or cannabinoids [154], estrogens [155], various drugs such as gliclazide [156], nadifloxacin [157], phenytoin [158], zolpidem [159], and pramipexole [160] among others. Finally, molecularly imprinted sorbents were also used to monitor in vivo in animal model metabolites of the pharmacologically active compounds of plant origin such as hesperetin, a flavonoid, possessing antioxidant, anti-cancer, anti-inflammatory, cardiovascular protection and anti-rheumatic activities [161].

To sum up, the molecularly imprinted sorbents were utilized in various analytical methods for the determination of environmental pollutants, food ingredients, or clinically important biomolecules. Their widespread application derived from high selectivity and satisfactory capability to clean up and remove interfering components of the samples.

## 6. Summary and Future Perspectives

Application of different types of molecularly or ion imprinted polymers as sorbents for analytical procedures received great attention during the final years. An unprecedented selectivity paired with high stability of imprinted material are the main advantages that allow use of imprinted sorbents during pretreatment procedures of samples with complicated matrices and with low levels of analytes. The simplicity and low cost of preparation enable the production of different forms of imprinted material appropriate for a selected extraction technique. Nevertheless, according to the extraction method, the sorbent should possess suitable properties characterized by appropriate techniques. The main problem of imprinted polymers is their heterogeneity with respect to the particle size and specific adsorption regions in the polymeric matrix. To solve these difficulties, an adequate optimization of synthetic processes is required. The polymerization technique is the most important example of such a process. Apart from the most common bulk polymerization method, other techniques that ensure the optimal shape and size of imprinted particles, such as in situ multi-step swelling, suspension/emulsion polymerization, surface molecular imprinting technique on chosen materials like mesoporous silica, nanofibers, or magnetic particles, often need to be applied. The application of new, specifically designed monomers and cross-linkers could also improve the performance of imprinted sorbents. Additionally, studies on the mechanisms of heterogeneity emergence in imprinted materials should be further carried out. We believe that, in the future, new imprinted materials will become widespread in different (well-known or new) analytical techniques during the analyses of biological, food, environmental, or pharmaceutical materials. These materials will be applied for construction of new devices for quantification and qualification of various compounds.

## Figures and Tables

**Figure 1 materials-14-01850-f001:**
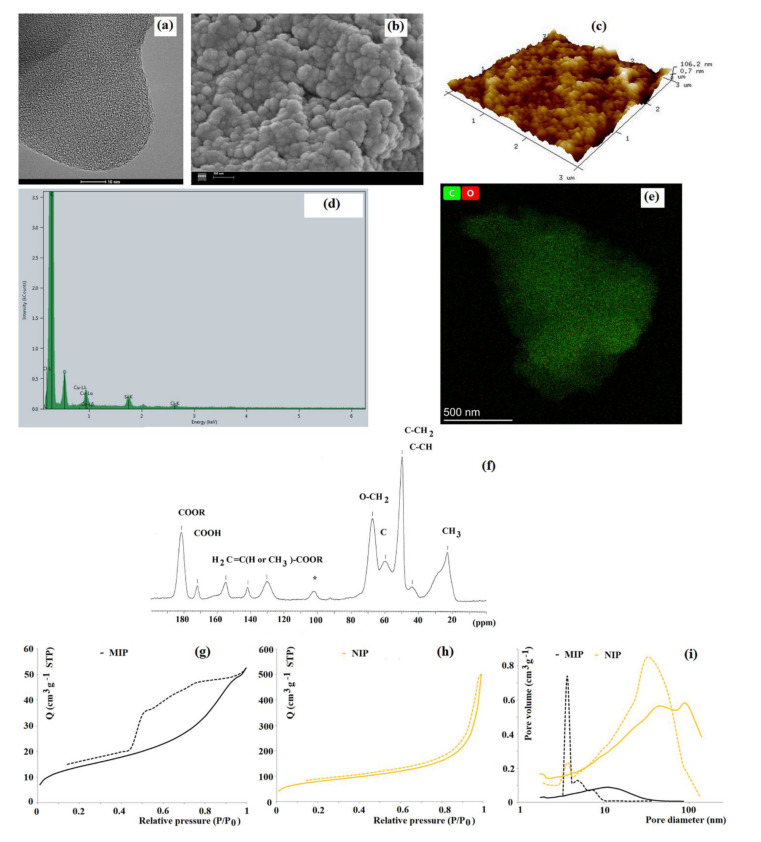
The transmission (**a**), scanning (**b**) electron, and atomic force (**c**) micrographs of imprinted sorbent together with elemental mapping from X-ray electron dispersive (**d**,**e**) and ^13^C CP/MAS NMR spectroscopies (**f**) as well as nitrogen sorption data (**g**–**i**) [30].

**Figure 2 materials-14-01850-f002:**
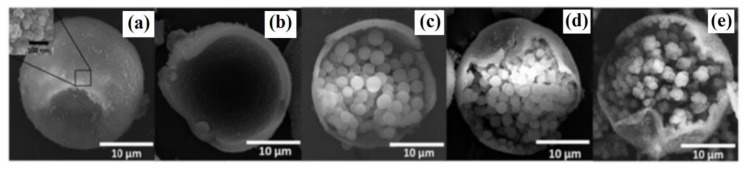
Morphology of microspheres: molecularly imprinted polymers (MIP) (**a**), non-imprinted polymer (NIP) (**b**), and MIP synthesized in various amounts of the template (**c**–**e**). Adapted with permission from Reference [76]. Elsevier, 2018.

**Figure 3 materials-14-01850-f003:**
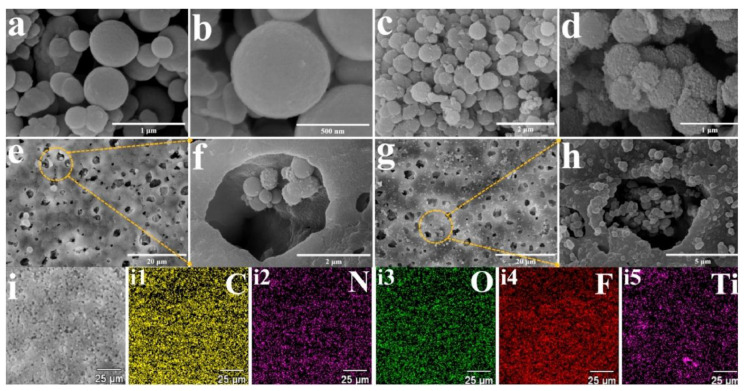
Scanning electron micrographs (**a**–**i**) and elemental mapping (**i1**–**i5**) for carbon, nitrogen, oxygen, fluor, and titanium) of imprinted nanocomposite based on TiO_2_ core and polydopamine shell. Adapted with permission from Reference [81]. Elsevier, 2020.

**Figure 4 materials-14-01850-f004:**
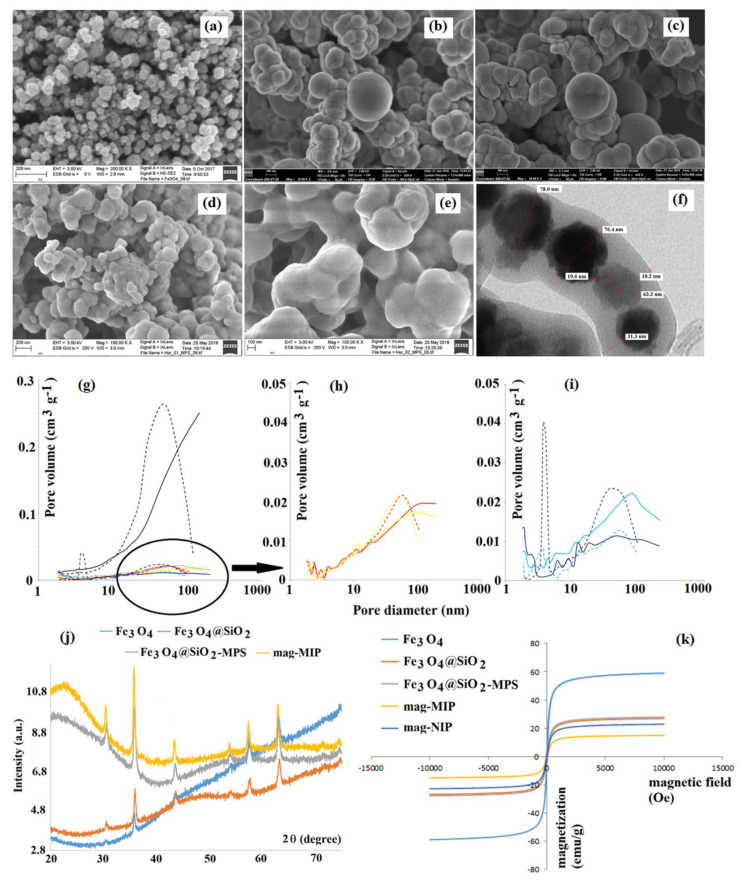
Scanning (**a**–**e**) and transmission (**f**) electron micrographs, nitrogen sorption data (**g**–**i**), X-ray diffractogram (**j**), and hysteresis from a vibrating sample magnetometer (**k**) of magnetic imprinted sorbent [84].

**Figure 5 materials-14-01850-f005:**

Three-dimensional distribution of poly-ε-caprolactone support from thin (red) to medium (green) and to thick (blue) for: unmodified (**a**), functionalized by a bismethacrylate polymer network (**b**), and final MIP material (**c**). Adapted with permission from Reference [94]. Elsevier, 2017.

**Figure 6 materials-14-01850-f006:**
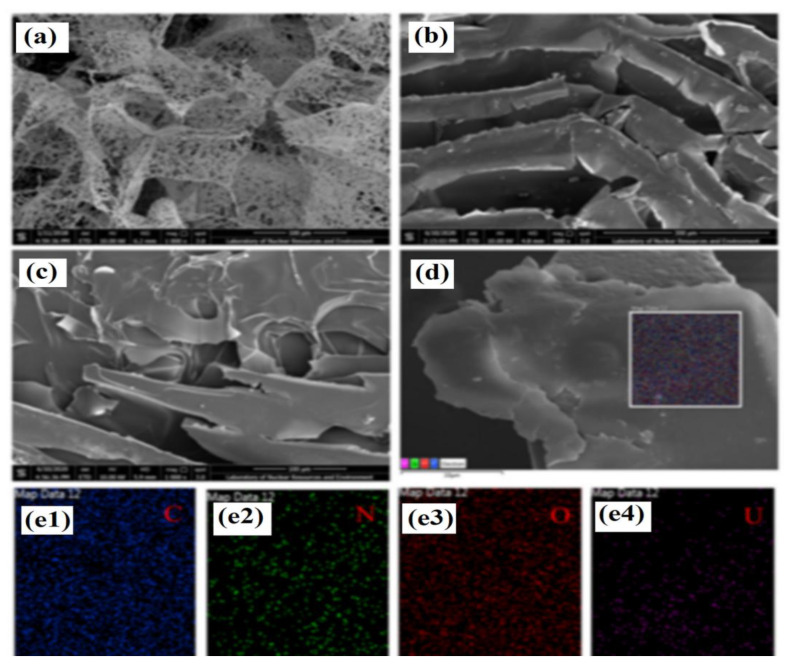
Scanning electron micrographs of ion imprinted (**a**,**c**,**d**) and non-imprinted (**b**) chitosan foam together with elemental mapping ((**e1**–**e4**), for carbon, nitrogen, oxygen, and uranium, represents material after loading with uranium ions). Adapted with permission from Reference [96]. Elsevier, 2020.

**Table 1 materials-14-01850-t001:** Summary of various forms of molecularly imprinted polymers (MIPs) together with their composition details, characterization methods, and application in the separation process.

MIP Form	MIP Matrix Components	Template	Analyte	Comments (Characterization, C; Application, A)	Ref.
Bulk polymer	Acrylic acid EGDMA	HNPA	HNPA	C: TEM, SEM, ^13^C CP MAS NMR, BET, XRD, ESD, FTIR A: HNPA extraction from artificial urine	[30]
Bulk polymer	MAA or MMA EGDMA	4,4′-dihydroxydiphenyl ether	PBDE-47, PBDE-99	C: BET, SEM, FTIR, TGA	[63]
Bulk polymer	4-vinylpyridine EGDMA	Naproxen and ketoprofen simultaneously	Naproxen, ketoprofen	C: HgP Co^2+^ was used as pivot during synthesis	[64]
Monolithic disc	MAA EGDMA	Ecgonine methyl ester	Ecgonine methyl ester	C: ATR-FTIR, SEM A: ecgonine methyl ester extraction from water samples	[65]
Mesoporous MIP	MAA EGDMA	7-acetoxy-4-methylcoumarin	Aflatoxins	C: SEM, EDS, XRD, ATR-FTIR Mesoporous silica FDU-12 was used as the carrier during synthesis, A: aflatoxins extraction from food samples	[67]
Mesoporous MIP	ICPTES TEOS	Bisphenol A	Bisphenol A	C: TGA, BET, SEM, FTIR, XPS Mesoporous silica SBA-15 was used as the carrier during synthesis, A: bisphenol A extraction from water samples	[68]
Mesoporous MIP	Alkyne-modified β-cyclodextrin and propargyl amine	2,4-D	2,4-D	C: BET, FTIR, XRD, TGA, elemental analysis, ^1^H NMR (for template study) Mesoporous silica SBA-15 was used as the carrier during synthesis, A: 2,4-D extraction from water samples	[69]
Precipitated MIP	4-vinylpyridine EGDMA	Cr (VI) ion	Cr (VI) ion	C: XRD, SEM, EDS, BET, FTIR A: adsorption of Cr(VI) from electroplating industrial waste	[70]
Precipitated MIP	MAA EGDMA	Tylosin tartrate	Tylosin tartrate	C: BET, DLS, SEM, FT-IR A: tylosin extraction from the broth	[71]
MIP microspheres	MAA TRIM	Efavirenz	Efavirenz	C: FTIR, SEM, DLS A: efavirenz extraction from urine and serum	[72]
MIP microspheres	MAA EGDMA	Alpha-(2,4-Dichlorophenyl)-1H-imidazole-1-ethanol	Climbazole, Clotrimazole Miconazole	C: BET, TGA, FTIR A: climbazole, clotrimazole, miconazole extraction from fish samples	[73]
MIP microspheres	4-vinylpyridine EGDMA	Bisphenol A	8 Bisphenols	C: BET SiO_2_ nanoparticles were used as the emulsion stabilizer during synthesis A: bisphenol extraction from urine	[74]
MIP microspheres	4-vinylpyridine EGDMA	Resveratrol	Resveratrol	C: FTIR, XRD, TGA, BET, ^1^H, ^13^C NMR (for template removal) Silanized porous cellulose microspheres were used as the carrier during synthesis A: resveratrol extraction from *Polygonum cuspidatum*	[75]
MIP multicore rattle-type microspheres	4-vinylpyridine EGDMA	Bisphenol A	Bisphenol A	C: SEM Silica nanoparticles were used as the emulsion stabilizer during synthesis	[76]
Core-shell MIP	Acrylamide EGDMA	Lincomycin	Lincomycin	C: DLS, SEM, FTIR Core: components: methacrylamide, EGDMA A: Lincomycin extraction from milk samples	[77]
Core-shell MIP	Dopamine	Lysozyme	Lysozyme	C: XPS, TGA, SEM, TEM, FTIR Core: carboxyl-functionalized carbon microspheres made from glucose and acrylic acid	[78]
Core-shell MIP	MAA EGDMA	Carbamazepine	Carbamazepine	C: SEM, FTIR Core: polystyrene-coated with siloxane and then polystyrene removal—siloxane shell A: carbamazepine extraction from water samples	[79]
Core-shell MIP	MAA EGDMA	Theanine	18 amino acids	C: SEM, TEM, FTIR Core: siloxane functionalized by VTMS A: separation of amino acids in tobacco and tobacco smoke	[80]
Core-shell MIP	MAA, acrylamide EGDMA	Artemisinin	Artemisinin	C: XPS, SEM, TEM, EDS (elemental mapping) Core: PDA-TiO_2_ nanoparticles functionalized by MPTS and poly(vinylidene fluoride)	[81]
Core-shell MIP	Aminophenylboric acid Dopamine	Luteolin	Luteolin	C: FTIR, XPS, DLS, SEM Core: siloxane coated with ZrO2 A: luteolin extraction from peanut shell samples	[82]
Core-shell MIP	MAA Tetraethoxysilane	S-amlodipine	S-amlodipine	C: BET, TGA, DSC, SEM Core: MOF-177	[83]
Magnetic core-shell MIP	MAA EGDMA	N,N-dimethylphenethylamine	Hordenine	C: SEM, TEM, VSM, XRD, BET Core: Fe_3_O_4_ coated with siloxane and functionalized by MPS A: hordenine extraction from plasma	[84]
Magnetic core-shell MIP	Acrylamide N,N-methylenebisacrylamide	Ce (III) ion	Ce (III) ion	C: TEM, TGA, BET, FTIR Core mesoporous SBA-15 material with Fe_3_O_4_ functionalized by RAFT agent A: Ce (III) ion removal from water samples	[85]
Magnetic core-shell MIP	Acrylamide EGDMA	3-phenoxybenzoic acid	Pyrethroids pesticides	C: BET, TGA, FTIR, VSM, SEM, TEM Core: NH_2_-functionalized Fe_3_O_4_ with graphene oxide A: pyrethroids pesticides extraction from fruit juices samples	[86]
Magnetic core-shell MIP	Phenyltrimethoxysilane Tetraethoxysilane	Aristolochic acid I	Aristolochic acid I	C: BET, TEM, XRD, FTIR, VSM Core: magnetic carbon nanotubes functionalized by carboxyl groups A: aristolochic acid I extraction from Traditional Chinese Medicine	[87]
Magnetic core-shell MIP	4-vinylphenylboronic acid	Luteolin	Luteolin	C: SEM, TEM, XPS, TGA, FTIR, BET, VSM Core: silica nano-bottles (NBs) capped with 3-chloropropyl groups and a hydrophilic interior surface (capped with amino-groups modified by succinic anhydride) and magnetic nanoparticles attached to carboxylic acid functionalized NBs	[88]
Magnetic core-shell MIP	Commercial	--	16 PAHs	Commercially available magnetic sorbent, provided by NanoMyP^®^	[89]
Magnetic stir bar MIP	MAA EGDMA	Propazine	6 triazines	Core: oleic acid functionalized Fe_3_O_4_ coated by silica A: triazines extraction from soil samples	[90]
Precipitated MIP	MAA TRIM	Metergoline	Metergoline	C: SEM, BET MIP was immobilized on three-dimensional printed scaffolds made from poly-ε-caprolactone	[94]
MIP microspheres	MAA TRIM	Benzyl paraben	3 Parabens	C: SEM MIP microspheres were entrapped in electro-spun polystyrene fibers A: parabens extraction from water samples	[95]
Imprinted microporous chitosan foam	Chitosan Epichlorohydrin	U (VI) ion	U (VI) ion	C: SEM, XPS, FTIR, EDS (elemental mapping)	[96]

ATR-FTIR—attenuated total reflectance Fourier transformed infra-red spectroscopy, BET—N_2_ sorption analysis, 2,4-D—2,4-dichlorophenoxyacetic acid, DSC—differential scanning calorimetry, DLS—dynamic light scattering, EDS—energy dispersive X-ray spectroscopy, EGDMA—ethylene glycol dimethacrylate, FTIR-Fourier transformed infra-red spectroscopy, HgP—mercury porosimetry, HNPA—4-hydroxy-3-nitrophenylacetic acid, ICPTES—3-(isocyanatopropyl)triethoxysilane, MAA—methacrylic acid, MMA—methyl metacrylate, MOF—metal-organic framework, MPS—3-(trimethoxysilyl)propyl methacrylate, MPTS—3-methacryloxypropyltrimethoxysilane, PAH—polycyclic aromatic hydrocarbon, PBDE—polybrominated diphenyl ether, PDA—polydopamine, RAFT—reversible addition-fragmentation chain transfer polymerization, SEM—scanning electron microscopy, TEM—transmission electron misroscopy, TGA—thermogravimetry, TEOS—tetraethyl orthosilicate, TRIM-trimethylolpropane trimethacrylate, VSM—vibrating sample magnetometer, VTMS—vinyltrimethoxysilan, XPS—X-ray photoelectron spectroscopy, XRD—powder X-ray diffraction.

## Data Availability

Not applicable—a review type paper.

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
