# Peer review of "Imprinting Technology for Effective Sorbent Fabrication: Current State-of-Art and Future Prospects"

_materials, 2021, doi:10.3390/ma14081850_

Round 1

Reviewer 1 Report

This review presents an exhaustive discussion on the advances in the imprinted polymers (bot molecule and ion) designed for road range of applications such as solid phase extraction, food analysis, biomedical analysis, environmental analysis etc. The manuscript is well structured and encloses a comprehensive overview of the imprinting process as well as four sections devoted to the synthesis and characterization of various kinds of imprinted sorbents for solid phase extraction. Possible applications future perspectives of imprinted sorbents are summarized in separate sections. The detailed discussion is supported by large number of relevant references.

Comments and suggestions:

  • In the manuscript the authors use word format when describing the form of the sorbent (e.g. line 17 : “Various formats such as bulk or monoliths, microspheres and core-shell...”; line 24: “obtained formats of MIPs and IIPs”; lines 97-98: “various synthetic approaches and material formats will be presented”; line 385: “various formats of monoliths were elaborated”; ). I would recommend the usage of such untypical term in this context to be reconsidered.
  • Figures in the manuscript are reproduced from cited references. In every caption it is stated “Reprint with permission” but the number of the corresponding reference is missing and should be included.
  • The title of chapter 6: “Current status and future perspectives” is confusing as the current status is actually covered by the entire content of the manuscript. I would recommend the title of this section to be revised.

Author Response

Response to Referees Comments:

We would like to thank all Reviewers for their valuable comments.

Referee #1:

We appreciate kind words related to our manuscript.

Ad. 1. In the manuscript the authors use word format when describing the form of the sorbent (e.g. line 17 : “Various formats such as bulk or monoliths, microspheres and core-shell...”; line 24: “obtained formats of MIPs and IIPs”; lines 97-98: “various synthetic approaches and material formats will be presented”; line 385: “various formats of monoliths were elaborated”). I would recommend the usage of such untypical term in this context to be reconsidered.

Thank you for suggestion. The word ‘format’ was removed from the text and in a few sentences it was substituted by the word ‘form’ (e.g. line 379 or 849).

Ad. 2. Figures in the manuscript are reproduced from cited references. In every caption it is stated “Reprint with permission” but the number of the corresponding reference is missing and should be included .

The citations were completed in the figure captions.

Ad. 3. The title of chapter 6: “Current status and future perspectives” is confusing as the current status is actually covered by the entire content of the manuscript. I would recommend the title of this section to be revised.

The heading of section was revised as follows: Summary and future perspectives.

Reviewer 2 Report

In this review manuscript, Janczura and co-workers have provided a general overview of the polymer imprinting technology applied to the development of new sorbents for analytical applications. The importance of molecularly imprinted polymers and ion imprinted polymers has been perfectly introduced and described, and a good overview of the imprinting process has been provided. Finally, different formats typically produced were also discussed. Despite all the previously mentioned, there are some aspects that should be considered and modified before considering this manuscript for publication.

  • The synthesis and application of MIPs and IIPs has been a topic of concern in the last years because of the advantages of these materials, especially in sample preparation approaches. As a consequence, many review articles have already been published in this sense. At least a short review of reviews section should be included after the introduction in order to provide a general overview of those review articles previously published. Besides, the period of time in which this review is focused on should be indicated.
  • An important aspect to be considered is the fact that tables have not been included. Tables summarizing and highlighting the most important aspects of the articles found in the literature provide valuable and clear information to the reader. This is extremely necessary in order to provide and demonstrate a suitable and complete revision of the literature.
  • In general, the descriptions of different research works are too long and detailed. This makes the reader become distracted, deviating from the main message that authors want to convey. This should be avoided including the tables mentioned in previous point, since the information included in those tables does not necessarily have to be discussed in the text. For example, the discussions about polymers characterization is suggested to be shortened.

Other specific comments:

  • Despite English is quite good, some minor grammatical mistakes should be corrected.
  • Keywords exceed the maximum allowed by the journal (10 keywords).
  • Line 21. Revise the sentence.
  • The quality of the Figures should be improved. In some cases they are difficult to read and details are impossible to distinguish.
  • Figure captions. The reference from which they are reprinted should be indicated.

Author Response

Response to Referees Comments:

We would like to thank all Reviewers for their valuable comments.

Referee #2:

We appreciate kind words related to our manuscript.

Ad. 1. The synthesis and application of MIPs and IIPs has been a topic of concern in the last years because of the advantages of these materials, especially in sample preparation approaches. As a consequence, many review articles have already been published in this sense. At least a short review of reviews section should be included after the introduction in order to provide a general overview of those review articles previously published. Besides, the period of time in which this review is focused on should be indicated.

The paragraph was dedicated to a brief review of reviews devoted to molecularly imprinted polymers. It was located in the last part of the Introduction. However, due to clarity of the manuscript the implementation of the additional section was omitted. The following text was added:

The excellent reviews summarizing the synthesis and application of MIPs and IIPs in various fields were published previously [8-17]. The most prominent reviews presented by BelBruno [9] as well as Chen and co-workers [10] describe the principles of the imprinting process and their applications. However, the limitations of the performance and handling of such materials are also emphasized. Cheong and co-workers [11] in an excellent review of reviews presented the development of imprinted sorbents in the previous decade. The progress in the application of MIPs as detectors resulted in a large number of reviews that were also published recently [12]. Nevertheless, the most interesting, but also the most challenging area of application of MIPs is related to protein separation, the detection of microorganisms and the usage of impritned materials in bioimaging and cell targeting [13-15]. Drug delivery vehicles based on the imprinted materials also attracted attention. A few different methods of application of MIPs for ocular, transdermal or oral administration were described in a review by Lulinski [16]. Finally, Murastugu and co-workers [17] revealed a potencial of MIPs as versatile tools for catalysis.

The references were revised and completed as follows:

  1. Chen, L.; Wang, X.; Lu, W.; Wu, X.; Li, J. Molecular imprinting: perspectives and application, Chem. Soc. Rev. 2016, 45, 2137-2211.
  2. Cheong, W.J.; Yang, S.H.; Ali, F. Molecular imprinted polymers for separation science: A review of reviews. J. Sep. Sci. 2013, 36, 609-628.
  3. Cui, B.; Liu, P.; Liu, X.; Liu, S.; Zhang, Z. Molecularly imprinted polymers for electrochemical detection and analysis: progress and perspectives. J. Mater. Res. Technol. 2020, 9, 12568-12584.
  4. Yarman, A.; Kurbanoglu, S.; Zebger, I.; Scheller, F.W. Simple and robust: The claims of protein sensing by molecularly imprinted polymers. Sens. Actuators B 2021, 330, 129369..
  5. Dar, K.K.; Shao, S.; Tan, T.; Lv, Y. Molecularly imprinted polymers for the selective recognition of microorganisms. Biotechnol. Adv. 2020, 45, 107640.
  6. Piletsky, S.; Canfarotta, F.; Poma, A.; Bossi, A.M.; Piletsky S. Molecularly imprinted polymers for cell recognition. Trends Biotechnol. 2020, 38, 368-387.

Additionally, a time-span of the current review was indicated as follows: In this review a critical revision of various synthetic approaches supported by comprehensive characterization is presented, covering the investigations published predominantly within the period of the last five years.

Ad. 2. An important aspect to be considered is the fact that tables have not been included. Tables summarizing and highlighting the most important aspects of the articles found in the literature provide valuable and clear information to the reader. This is extremely necessary in order to provide and demonstrate a suitable and complete revision of the literature.

Table entitled: Forms of MIPs together with their composition details and application in separation process, was prepared and inserted into the text (line 903).

Ad. 3. In general, the descriptions of different research works are too long and detailed. This makes the reader become distracted, deviating from the main message that authors want to convey. This should be avoided including the tables mentioned in previous point, since the information included in those tables does not necessarily have to be discussed in the text. For example, the discussions about polymers characterization is suggested to be shortened.

The text was revised and parts of the manuscript, starting from lines 330, 536, 562, 577, 607,  825, and 837 were shortened and re-written. Those fragments were red-indicated.

Ad. 4. Despite English is quite good, some minor grammatical mistakes should be corrected.

English language was revised by English-speaking specialist.

Ad. 5. Keywords exceed the maximum allowed by the journal (10 keywords).

The list of keywords was reduced to eight items. The following keywords were deleted: food analysis, biomedical analysis, environmental analysis.

Ad. 6. Line 21. Revise the sentence.

Sentence in line 21 was revised, shortened and combined with the next sentence as follows: Bulk or monoliths, microspheres and core-shell materials, magnetic susceptible and stir-bar imprinted materials are applicable to different modes of solid-phase extraction to determine target analytes and ions in very complex environment such as blood, urine, soil or food.

Ad. 7. The quality of the Figures should be improved. In some cases they are difficult to read and details are impossible to distinguish.

The quality of Figure 1 and 4 was improved.

Ad. 8. Figure captions. The reference from which they are reprinted should be indicated.

The citations were completed in the figure captions.

Round 2

Reviewer 2 Report

After reviewing the revised version of the manuscript, it can be seen that the authors have made important changes that have improved the overall quality of the review. However, it should be mentioned that many changes have been made that have not been marked, which makes it difficult to appreciate.

I suggest modifying the table, including some information related to the characterization of the MIPs that allow to simplify some parts of the text as I indicated before. The manuscript is still overly detailed in parts, which causes the reader to be distracted.

Other specific comments:

Line 112: Imprinted.

Line 901: Table 1 summarizes…

Table 1: A change in the title of the fifth column (application, comments…) is suggested.

Author Response

Response to Referees Comments:

We would like to thank all Reviewers for their valuable comments and detail revision.

Referee #1:

Ad. 1. However, it should be mentioned that many changes have been made that have not been marked, which makes it difficult to appreciate.

In the current version of the manuscript all fragments of the text which were deleted or inserted are clearly indicated.

Ad. 2. I suggest modifying the table, including some information related to the characterization of the MIPs that allow to simplify some parts of the text as I indicated before.

The Table 1 was modified according to Referee suggestion. The fifth column was devoted to ‘Comments’, including characterization methods and application. The legend was extended for abbreviations of methods.

Ad. 3. The manuscript is still overly detailed in parts, which causes the reader to be distracted.

The Section 3 devoted to synthetic approaches and characterization of imprinted sorbents was substantially reduced. The characterization methods were placed in the Table 1 and extensive characterization data were deleted. However, due to the fact that the manuscript belongs to the Special Issue entitles ‘Synthesis and characterization of imprinted sorbents’, in the selected descriptions, characterization data was remained to discuss analysis of imprinted sorbent in details. Additionally, the brief text was added into the beginning of the Section 3:

In this section, synthetic approaches to fabricate molecularly or ion imprinted sorbents will be discussed together with characterization of obtained materials. Various forms of MIPs together with their composition details, characterization methods and application in separation process are summarized in Table 1 at the end of section.

Other specific comments:

Line 112: Imprinted.

The error was corrected.

Line 901: Table 1 summarizes…

The Table caption was corrected.

Table 1: A change in the title of the fifth column (application, comments…) is suggested.

The title of column was changed to: Comments (Characterization, C; Application, A).

We hope that the corrections will satisfy the Referees and that the current revised version of our manuscript will be valuable paper and proper to be published in this journal.